# *Helianthus annuus* Seed Extract Affects Weight and Body Composition of Healthy Obese Adults during 12 Weeks of Consumption: A Randomized, Double-Blind, Placebo-Controlled Pilot Study

**DOI:** 10.3390/nu11051080

**Published:** 2019-05-15

**Authors:** Aurélie Leverrier, David Daguet, Wim Calame, Pierre Dhoye, Shyam Prasad Kodimule

**Affiliations:** 1Vidya Europe SAS, 7 avenue de Norvège, 91140 Villebon sur Yvette, France; aurelie@vidyaeurope.eu (A.L.); pierre@vidyaeurope.eu (P.D.); shyamprasad@vidyaherbs.com (S.P.K.); 2StatistiCal BV, Strandwal 148, 2241 MN Wassenaar, The Netherlands; w.calame@kpnplanet.nl; 3R&D Centre for Excellence, Vidya Herbs Pvt. Ltd., #14A, Jigani I Phase, Bangalore 560105, Karnataka, India

**Keywords:** sunflower seeds, chlorogenic acids, weight loss, BMI reduction, fat mass reduction, cholesterol

## Abstract

The aim of this pilot study was to evaluate the effects of a sunflower (*Helianthus annuus*) seed extract, standardized for 40% chlorogenic acids on weight and body composition of obese adults. Fifty subjects were randomly assigned to sunflower extract or isocaloric placebo groups, receiving respectively 500 mg/day of treatment for 12 weeks. At the end of the intervention, a significant decrease in body weight, Body Mass Index (BMI), and waist circumference was observed, especially for obese female subjects above 30 years. Those changes were associated with modified body composition related to fat mass loss. A decrease in blood cholesterol was also observed, supporting the potential action of sunflower extract on lipid metabolism. It was concluded that consumption of sunflower extract has a beneficial effect on body weight, fat mass, and lipid profile, providing evidence for its use as a natural anti-obesity herbal extract.

## 1. Introduction

Overweight and obesity are defined as abnormal or excessive fat accumulation that may impair health. In the last decades, obesity has become a serious growing and global problem, because of a constant increase of the population concerned, which nearly tripled between 1975 and 2016 [1]. According to the World Health Organization (WHO), in 2016, 39% of the global adult population was overweight (Body Mass Index (BMI): 25–30) and 13% obese (BMI ≥ 30). Obesity is a strong risk factor for various diseases like diabetes, hypertension, cardiovascular diseases, arthritis, atherosclerosis, and certain types of cancer, and it has been associated with the prevalence of comorbid illnesses, health complaints, and physical disabilities [2]. According to the WHO, in 2014, obesity was responsible for 44%, 23%, and between 7% and 41% of the cases of diabetes, ischemic heart diseases, and certain cancers, respectively [3].

Nowadays, the treatment of obesity and overweight mainly relies on diet, physical activity, lifestyle change, and behavior therapy, and in some case, have resorted to surgery or pharmacotherapy for high-risk patients. Despite the important number of studies in the field of anti-obesity pharmacotherapy, most of the synthetic molecules failed to reach the market or were withdrawn, because of their hazardous side-effects, such as nausea, headache, dizziness, and gastro-intestinal disorders [4,5]. Those observations, associated with the high cost of such drugs, led to the consideration of plant-based products, as safe and low-cost alternative strategy for the development of anti-obesity products. Numerous herbal extracts and natural compounds were indeed reported for their positive effects on weight management and metabolic disorders [6,7].

Sunflower (*Helianthus annuus*, Asteraceae) is an oleaginous plant native from North America and today grown worldwide as a crop primarily for its edible seeds. Sunflower seed contains 35–42% oil, and oil production from sunflower seeds is ranks fourth in the world, representing 8% of the total global oil production in 2012 [8]. Sunflower seeds contain valuable nutrients, proteins, vitamins, minerals, and antioxidants. In particular, phenolic compounds account for 1–4% of the total mass of sunflower residues from oil extraction, chlorogenic acids (CGAs) being the major components [9].

Chlorogenic acids are a class of polyphenol compounds, formed by esterification of cinnamic acid with (-)-quinic acid (QA), abundantly found in human diet. They are produced by many plants, and green coffee is one of the major sources found in nature (5–12 g/100 g) [10]. The health benefits of CGAs were mostly associated with their antioxidant and anti-inflammatory activities. In particular, CGAs were reported to have hypolipidemic, hypoglycaemic, and antidiabetic effects by regulating glucose and lipid metabolisms [11]. In-vitro and in-vivo studies reported that CGAs could improve glucose tolerance, stimulate insulin secretion, improve insulin resistance, reduce postprandial blood glucose levels, and improve lipid profile by lowering serum and hepatic triglycerides levels, reducing low density lipoproteins (LDL) oxidation susceptibility, decreasing LDL cholesterol level, inhibiting fat absorption, and activating fat metabolism [11].

We therefore hypothesized that sunflower seeds could be a sustainable source of CGAs with potential anti-obesity properties. Notably, a recent in-vivo study reported the safety and the antidiabetic effect of an ethanolic sunflower seed extract [12]. Saini and Sharma observed a decrease of blood glucose level, an increase in liver glycogen, and a decrease of cholesterol in diabetic rats compared to control rats, which was explicated by an increase of insulin secretion [12]. In the present study, we investigated, for the first time, the effect of a sunflower seed extract standardized for CGAs on body weight and body fat reduction in obese adults, compared to an isocaloric placebo. The aim of the present work was to explore the clinical efficacy and the safety of this sunflower extract as an anti-obesity product.

## 2. Materials and Methods

### 2.1. Study Design

A randomized, placebo-controlled, double-blind, parallel group clinical pilot study was conducted over 12 weeks to evaluate the effects of a dietary supplementation with sunflower extract on body weight, anthropometric, and biochemical parameters of obese adults. The study was carried out at the Clinical Trials Unit (CTU) managed by Tecnalia Research & Innovation, situated in Araba University Hospital-Site Txagorritxu, Vitoria (Alava, Spain). Volunteers were recruited by advertisements and enrolled in the randomization when conformed with inclusion/exclusion criteria. All study participants gave written informed consent to take part in the clinical trial. The study was conducted in accordance with both national and international regulations (International Conference on Harmonization (ICH) Guidelines), respecting the principles of the declaration of Helsinki, and after the approval from the Research Ethics Committee (REC) of Araba University Hospital, Vitoria-Gasteiz. This clinical trial has been authorized by the Basque Country sanitary authorities, Osakidetza, at the date of 20 January 2016 and registered as Vidya Europe SAS (Expte.2016-98).

### 2.2. Subjects

A total of 66 volunteers of both sexes, being 18–65 years old, BMI between 30 and 40 kg/m^2^, and waist circumference greater than 102 cm for men and 88 cm for women, were screened during the selection visit (V0), in order to evaluate their suitability for the trial. Excluded were individuals having or suspected of having medical conditions that prevent them from participate to the trial, such as acute or chronic gastro-intestinal or metabolic pathologies, liver, heart, and kidney diseases, or severe hyper-tension (Systolic Blood Pressure (SBP) > 120 mm); heavy drinker and/or smokers; individuals operated on bariatric surgery; individuals having thyroid disorders; and subjects under weight loss medication or diet supplement; individuals taking a corticoid treatment; or treatment with significant changes in dose/type in the three months prior inclusion. Pregnant and post-menopausal women, and women taking contraceptives with significant changes in dose/type for three months prior to inclusion were also excluded. Finally, individuals having history of allergy to sunflower seeds, nuts, pollen, or vegetable pan allergens were excluded, along with people who took part in another clinical trial during the two months prior the trial. 

Among the 66 volunteers screened, 50 were randomized using MAS program version 2.1 (Design C4 Study Pack, Department of Biometrics/Medical of GlaxoSmithKline) and assigned to receive sunflower extract (30 subjects) or placebo (20 subjects).

### 2.3. Intervention Treatments and Composition of the Interventional Product

Participants were instructed to follow a hypocaloric diet (reduction of their energy intake by 500 kcal/day) given at V1 and monitored at visits V2, V3, V4, and V5 by a 24 h reminder questionnaire, and to take 2 capsules/day, containing either sunflower extract or placebo/day (one before breakfast and one before lunch) for 12 weeks. Each capsule contains 250 mg of sunflower extract, for active group, or 250 mg of maltodextrin (DE 19, Prochimia, Vallet, France), for placebo group, the capsules being identical in appearance and calories (374 and 385 kcal/100 g respectively). Sunflower extract (SUNCA^TM^) was supplied by Vidya Herbs Pvt. Ltd. (Bangalore, Karnataka, India) and consist of a hydroalcoholic extract of sunflower seed standardized for chlorogenic acids. High Performance Liquid Chromatography (HPLC) analysis assessed the presence of 6 major chlorogenic acids, namely 3-CQA, 5-CQA, 4-CQA, 3,4-Di-CQA, 3,5-DiCQA and 4,5-Di CQA (Figure 1), which were quantified as 5-CQA equivalent. HPLC analysis of the sunflower extract was carried out using a Shimadzu LC Nexera XR system, equipped with a quaternary pump and a DAD detector (SPD-M20A, Noisiel, France). The HPLC method used herein was adapted from methods published by Aqeel et al. [13] and Gouthamchandra et al. [14]. Briefly, a solution of the sunflower extract was prepared in methanol (at about 2 mg/mL) (UPLC grade, Panréac, Barcelona, Spain) and 10 µL were injected into a C-18 Kinetex column (2.6 µm, 4.6 × 150 mm, 100 Å, Phenomenex), eluted with a gradient of formic acid 0.1% in water (A) (HPLC grade, Sigma-Aldrich, St. Quentin Fallavier, France) and acetonitrile (B) (UPLC grade, Panréac, Barcelona, Spain), from A:B 95:5 to 80:20 in 9 min, maintained 3 min, from A:B 80:20 to 100% B in 0.5 min, maintained 2.5 min, followed by return to start conditions. Flow rate was maintained at 1.5 mL/min and CGAs were detected at 330 nm. The 6 major chlorogenic acids were identified according to results previously published by one of us (SPK) [14] and quantified as 5-CQA acid (tr = 5.3 min) from their peak areas, using a 6-points calibration curve (R^2^ = 0.999) prepared with 5-CQA standard (95%, Sigma-Aldrich, St. Quentin Fallavier, France) (solutions in methanol from 250 to 4 µg/mL).

Table 1 gives the HPLC retention times and the % (*w*/*w*) of CGAs which account for 44.6% of the extract, 5-CQA being the major compound (26.7% *w*/*w*) and accounts for nearly 60% of total CGAs. The extract composition is in accordance with CGAs previously described in sunflower seeds [9]. 

### 2.4. Study Protocol

During the screening visit (V0), clinical histories were recorded, consent forms were signed and biochemical, anthropometric, and safety parameters were measured to be used as baseline. The 50 subjects were then enrolled in a 12 weeks follow-up intervention, during which 5 visits separated by 3 weeks were planned. Participants were randomized in the first study visit (V1) and received their treatment for the 3 following weeks, their hypocaloric diet recommendation (based on their 24 h intake) and physical activity was recorded on an International Physical Activity Questionnaire (IPAQ). The hypocaloric recommendations were provided to the subjects through a 24 h reminder questionnaire, for a reduction of their energy intake by 500 kcal/day from the amount needed (energy expenditure—500 kcal/day) to maintain a stable weight. During experimental visits V2, V3, V4 and V5, vital signs and anthropometric measurement were determined, and blood samples were collected. Participants received study treatment for the next 3 weeks, empty treatments were picked up and the 24 h reminder diet recommendations were checked. Adverse effects and concomitant medications were also recorded. For the last visit (V5), 7-day physical activity recall was besides recorded on an IPAQ and status of the participants were evaluated by physical examination, analysis of blood and urine, and electrocardiogram (ECG). A summary of the study follow-up is presented in Appendix A.

### 2.5. Primary/Secondary Outcomes

The primary outcomes of the present pilot study were the effect of sunflower extract supplementation on weight loss and improvement of anthropometric measurements (such as fat loss) of obese subjects. The secondary outcomes were the improvement of lipid and glycaemic profiles and the assessment of the safety through clinical (vital signs and systemic adverse effects) and biological (blood count and biochemistry) monitoring.

### 2.6. Anthropometric Measurements

During visits V0 and V2–V5, subjects were monitored for anthropometric parameters (waist circumference (cm) (WC), body weight (kg), height (m) and BMI (calculated as weight in kg divided by square of height in meters)) and for their body composition evaluated by digital impedance monitor (Fat %, fat mass (FM), lean mass (MM), MM/FM ratio and total body water (TBW) (Tanita, Body Composition Analyzer, TBF-300 and SC-24OMA for weight and bio-impedance analysis, Tanita Europe BV, The Netherlands)).

### 2.7. Blood Sampling and Biochemical Analysis

For the evaluation of the main analytical variables, a blood extraction has been performed. Samples were collected in vacuum tubes with anticoagulant for serum extraction. Subsequently they were stored at 4 °C until to be sent to the biochemical laboratory for analysis. Fasting blood samples were collected at visits V0, V2–V5 to evaluate the efficacy and safety analytical variables. Analysis included glucose (mg/dL), Glycated hemoglobin HbA1c (%), Total Cholesterol (mg/dL), High density lipoproteins (HDL) (mg/dL), LDL (mg/dL), triglycerides (mg/dL), and fee fatty acids (FFA) (nmol/dL) were performed to evaluate the effect of the sunflower extract on lipid and glycemic blood profiles, at all the experimental visits.

Safety analysis also included hematology and urine analysis, performed at V0 and V5.

### 2.8. Safety Assessment

Vital signs (systolic and diastolic blood pressure, systolic blood pressure (SBP), diastolic blood pressure (DBP), pulse rate, and body temperature) were measured at screening visit (V0) and at visits V2–V5, by medical staff. Electrocardiogram (ECG) was recorded at both the screening (V0) and final (V5) visits (GE Medical Systems, Mac 1200 ST, Schiller, Bochum, Germany).

Possible adverse events (AEs) and concomitant medications were recorded during the study. AEs were listed according to their classification as serious and non-serious together with all requested items, e.g., duration, severity, outcome, and causality assessment. Incidences of the occurrences have been evaluated and categorized.

### 2.9. Statistical Analysis

Firstly, a comparison of the baseline variables in the two groups was performed using an unpaired t-test, when the normality of the distribution of the data was assessed through the Shapiro-Wilk test, or the Wilcoxon signed rank test, when normality was rejected. Chi square analysis was conducted to evaluate the differences in discrete outcome at the start, such as gender distribution. From all values, obtained at the various intervals of the study, the outcome at the start (V0) was subtracted. This yielded change values per person with the start value at “0”. These values were used in the statistical analysis as described below.

Subsequently, repeated measures through the General Estimating Equations (GEE) model was constructed, by applying a dummy approach in a stepwise fashion. The model included various confounding factors, such as age, gender, and start value of the respective dependent parameter, but most importantly time (as monitored via the various visits), treatment (being the dummy parameter: 0: placebo and 1: sunflower extract) and the interaction between time and treatment. During the stepwise procedure, the normality in the residual fraction was automatically checked. Only when the model showed a significant outcome, as evaluated via Wald Chi square test, the significance of the remaining (independent) parameters indicated that the observed combination could explain the outcome reached for the respective parameter in the present study. During the procedure, both the normality in the residual fraction, as well as the explained variation (by the model), were checked.

The GEE model was also applied in order to estimate the parameters with possible correlation between dependent outcomes. The model included various confounding factors, such as age, gender, start time, time, treatment, and the interaction between time and treatment.

Throughout the manuscript, data are expressed as mean ± standard error of the mean/standard deviation (s.e.m./SD) of the change values obtained from each subject. Throughout the current study, a *p*-value below 0.05 was considered to represent statistical significance applying a two-sided evaluation. Statistical analyses were performed using STATA, version 12.1 (StatCorp, College Station, TX, USA) and GraphPad, version 6 (GraphPad Prism, LaJolla, CA, USA), the latter was also applied for graphical display.

## 3. Results

### 3.1. Study Flow, Participants Characteristics and Safety

The design of the trial is depicted in Figure 2. A total of 66 subjects were examined during the screening visit and 13 were excluded as they did not meet the inclusion criteria: Four subjects had a BMI out of criteria, 3 subjects were smokers, 1 subject did not show analysis results, 4 women were post-menopausal, and 1 subject had chronic gastro-intestinal pathology. After completing the selection process, two suitable subjects refused to take part in the trial for reasons not associated with the current experiment and another one was post-visit excluded because of screening failure. A total of 50 subjects were therefore randomized and allocated to sunflower extract (30 subjects) or placebo (20 subjects) group. Among the 50 subjects, 46 subjects completed the 12-weeks study, 4 subjects gave up at V2 or V3, 2 subjects in each group, for personal reasons.

Table 2 details the baseline descriptive values of the parameters evaluated during the screening visit (V0) and shows that there were no significant differences between the two groups in terms of baseline values for anthropometric nor biochemical measures (*p* > 0.05). In both groups, the majority of participants were women, 75.0% of placebo group and 56.6% of sunflower extract group, being not significantly different as checked via Chi square. No significant differences were found for physical activity at baseline and at the end of the study, within, and between the two groups, according to the IPAQ results (*p* > 0.05).

The intervention treatments were well tolerated. The most common adverse events occurring in the study were headache and constipation. For headache adverse events, the rate of sunflower extract/placebo was 10/10, and for constipation 11/5, but without leading to withdrawal from the study or giving-up. No significant alterations in the vital signs, safety biochemical analysis, neither in the physical examination, nor in ECG between the two groups was observed.

### 3.2. Effect of Sunflower Extract Supplementation on Anthropometric and Biochemical Parameters

Table 3 gives the mean change values between V5 and V0. Comparisons between groups revealed that there are significant beneficial time-dependent changes (*p* < 0.05) for most of the parameters, before and after the intervention in both groups, which suggests good adherence of the subjects to the protocol. 

Comparison between groups revealed that sunflower extract consumption for 12 weeks significantly reduced BMI and waist circumference by −2.60, and −8.44 cm, respectively, compared to the placebo (reduction by −1.88 and −4.75 cm, *p* = 0.02, and 0.001, respectively). At the same interval, body weight was also reduced by 6.90 kg for sunflower extract group against 5.53 kg for the placebo group, consisting of a 6.8 and a 5.7% reduction, respectively, though not significantly. 

Significant changes in body composition were also observed in both groups (*p* < 0.05). Comparison between group revealed that changes in MM, FM, TBW and Fat % tended to be higher for subjects who took sunflower extract, compared to placebo, although not significant.

Finally, an improvement of lipid profile (reduction in both cholesterol as well as in LDL) was observed within both groups. Significant difference between sunflower extract and placebo was only detected for cholesterol values with reduction by −18.43 mg/dL for sunflower extract group against −8.72 mg/dL for placebo group (*p* = 0.02) at 12 weeks. However, the LDL data suggested a longer-lasting reduction after 12 weeks in sunflower extract group compared to placebo group (Figure 3), which suggests that sunflower extract tends to have a higher impact on lipid profile than placebo. A summary of the changes of parameters per visit compared to baseline for all the subjects is presented in Appendix A.

### 3.3. Optimization via Age and Sexes

Statistical analysis of the associations between variables, revealed that age had a significant impact on weight loss; the older the person, the larger the reduction in weight (*p* < 0.002). Given the few numbers of participants under 30 years old (4 in total, 2 in each group), a re-evaluation of the parameter changes was undertaken, with subjects > 30 years without a major loss in power. A summary of the changes of parameters per visit compared to the baseline of subjects above 30 years old is presented in Appendix A. In the same way, an association was found between body composition and gender. Given the known different distribution in fat/lean mass between genders, and the large proportion of women in the cohort, the results were then re-evaluated in the sub-group of women > 30 years. Mean changes of the parameters between V5 and V0 for subjects > 30 years and women > 30 years are depicted in Table 3.

When considering subjects > 30 years, besides BMI and WC, a significant difference was found between the two groups for weight loss (*p* = 0.043); subjects allocated to sunflower extract lost 7.28 kg against 5.71 kg for placebo group, i.e. a reduction of −7.2%, and −5.9% of initial weight, respectively. Moreover, significant differences were observed in the reduction of fat mass and fat %: −5.26 kg and −2.69% for sunflower extract group against −4.02 kg and −2.00% for placebo group, respectively (*p* = 0.018 and 0.047, respectively). For women above 30 years old, a significant improvement in MM/FM ratio was observed in the sunflower extract group (+0.14) compared to placebo group (+0.09) (*p* = 0.004). A summary of the changes of parameters per visit compared to the baseline of women above 30 years old is presented in Appendix A.

### 3.4. Associated Parameters

The association between body weight changes and other parameters was studied using a multivariate regression and GEE on the whole population of participants. Not unexpectedly, the changes, in fat mass and in lean mass were highly correlated with that in total weight. Fat mass was especially found to be strongly correlated (Figure 4). More than in lean mass, the model (Wald Chi square 1424.75; *p* < 0.0001) indicated that the change in weight is strongly associated with that in fat mass: *p* < 0.001 (adjusted R^2^: 0.83), for all the subjects. When considering the subgroup of women (Figure 4), association with loss in fat mass is even stronger with adjusted R^2^ = 0.87 (Wald Chi square 1244.39; *p* < 0.0001).

## 4. Discussion

The primary objective of the present study, which, to our knowledge, constitutes the first Randomized Clinical Trial (RCT) involving CGAs from sunflower, was to investigate the effect of a sunflower seed extract standardized for > 40% CGAs, on the anthropometric and biochemical parameters of obese adults. The findings showed that the consumption of 500 mg/day of a 40% chlorogenic acids sunflower extract for 12 weeks, significantly decreases BMI, waist circumference, body weight, and improves fat-related parameters, more specifically in women above 30 years.

Firstly, the results showed that, in the overall population evaluated, the consumption of this sunflower extract compared to that of placebo, significantly lowers BMI, by −2.60 against −1.88, and waist circumference (WC), by −8.44 against −4.75 cm. Women above 30 years particularly respond to sunflower extract consumption, and lost 9.15 cm WC on average against 3.23 cm for placebo group, i.e., almost three times more. The International Day for the Evaluation of Abdominal Obesity (IDEA) study looked at these two measures of fatness, WC and BMI, and found that they were strongly correlated with cardiovascular diseases and diabetes mellitus [15]. In the same way, it has been shown that a 5 to 10% reduction in initial weight was associated with a decrease in the risk factors associated with obesity and being overweight [16]. A decrease in body weight was observed during this study, in particular for subjects above 30 years who lost 7.7% of their initial weight when consuming the sunflower extract. Those observations support the potential benefits of this extract against cardiovascular disease- and obesity-associated risk factors.

Moreover, CGAs, the active components of the sunflower extract, were widely studied for their anti-obesity effects and were the subject of several clinical trials [11,17]. Our results are in accordance with a previous study, reporting a significant reduction of 4.97 kg (−5.7%) body weight and of −1.9 BMI after 60 days consumption of 400 mg/day of CGA enriched green coffee extract, compared to the placebo [18]. Dellalibera et al. also reported a significant increase of the MM/FM ratio, and explained their results by an increase in the consumption of fatty acid deposits, and by preventing them from being accumulated, which would be in part due to an inhibition of glucose absorption in the small intestine and inhibition of the activity of glucose-6-phosphatase [18]. In this study, a significant loss of fat mass was similarly observed for all subjects above 30 years, along with a significant improvement in the lean-to-fat mass ratio for women above 30 years of age (+0.14 for sunflower extract group against +0.09 for placebo group). Moreover, statistical work, using multivariate regression and GEE, proved that a change in body weight was strongly associated with change in fat mass, in both men and women. The loss of body weight is therefore explained by a loss of fat mass, in particular, for women above 30 years, the number of men involved in this study being too small to conclude with this particular sub-group. Moreover, the gender-specific fat reduction efficacy by sunflower extract might be explained by the difference in fat distribution between men and women, with men having a higher percentage of visceral fat and women that of subcutaneous fat. Metabolic responsiveness might differ between both compartments upon the intake of the investigational product. These results support the potential regulation of lipids metabolism by the sunflower extract. Animal studies previously reported that CGAs induce a reduction in body weight gain and visceral fat in obese mice, explained by an inhibition of fat absorption and accumulation associated with an activation of fat metabolism [11,19]. In the same way, we evaluated this sunflower extract in vivo, and found that, it decreased fat and brown adipose tissue, in high-fat-diet-fed rats by activation of liver Adenosine monophosphate-activated protein kinase (AMPK) pathway in a dose-dependent manner (50, 100, and 150 mg/kg) (data not shown). In addition, internal in vitro studies (data not presented) showed that this extract inhibited lipid accumulation in adipocytes and interacted with (Peroxisome proliferator-activated receptor) PPARα and γ, as it has been already reported for CGAs [11]. This sunflower extract would therefore regulate metabolism of lipids and energy balance through AMPK pathway, and regulate adipogenesis through the interaction with PPARα and γ. We allocated the weight loss properties of sunflower seed extract to the presence of CGAs, which may have a dose-dependent action, as suggested by animal studies and previous clinical trials.

Finally, the reduction in cholesterol, LDL-cholesterol, and triglycerides (TG) blood levels were found higher in sunflower extract group compared to placebo group, though mostly not significantly. Previous animal studies already reported a dose-dependent improvement of lipid profile after consumption of CGAs [19,20]. For example, Wan et al. observed a reduction of serum cholesterol and LDL in hypercholesterolemic rats after the administration of 10 mg CGA/kg for 28 days (equivalent to 1.6 mg/kg in human) [20]. Shimoda et al. observed a strong reduction of serum TG in mice after administration of 400 mg/kg of green coffee bean extract standardized for 27% CGA (108 mg CGAs/kg) [19]. This would be equivalent to 8.8 mg CGAs/kg in human, which would represent an administration of about 1800 mg of the sunflower extract for a 90 kg subject. This is consistent with our observation on mice (data not presented) and suggests that a higher dose of this extract could have a greater impact on the lipid profile. Regarding the changes over the five visits, the time dependent reduction of cholesterol was found larger after consumption of sunflower extract than that of placebo (*p* < 0.02). The data suggest that a longer treatment duration would also have shown significant reduction of these blood parameters (cholesterol, TG, and LDL). Consequently, due to the atherogenic properties of hyperlipidemia, including high levels of cholesterol and triglycerides that are known as important risk factors for cardiovascular diseases [21], supplementation with sunflower CGA extracts may contribute to the prevention or reduction of obesity-associated cardiovascular diseases.

## 5. Conclusions

In conclusion, despite the few numbers of subjects involved in this pilot study, the outcomes are encouraging in considering chlorogenic acids enriched sunflower extract as a natural anti-obesity treatment. Our results showed that the consumption of sunflower CGA extract (500 mg/day) for 12 weeks contributes to reduce BMI, waist circumference and body weight, mainly explained by a loss of mass in the fat compartment, more than that in the muscular compartment. Those changes may be accompanied by an improvement of lipid profile, which constitutes a current approach in the prevention of cardiovascular diseases and associated obesity diseases, like type 2 diabetes and atherosclerosis. A larger and longer study, involving more subjects of both sexes, is nevertheless warranted to more precisely assess the benefits of this sunflower extract.

## Figures and Tables

**Figure 1 nutrients-11-01080-f001:**
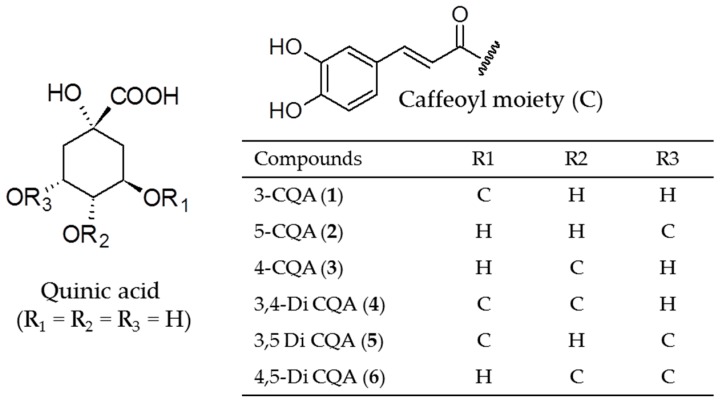
Structures of chlorogenic acids in sunflower extract.

**Figure 2 nutrients-11-01080-f002:**
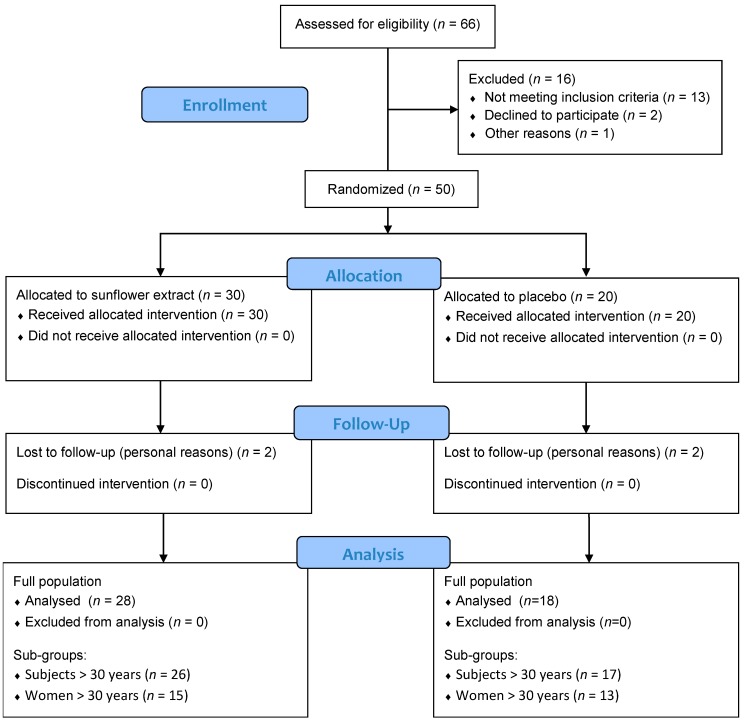
Diagram of the study design and completion of 12-weeks evaluations.

**Figure 3 nutrients-11-01080-f003:**
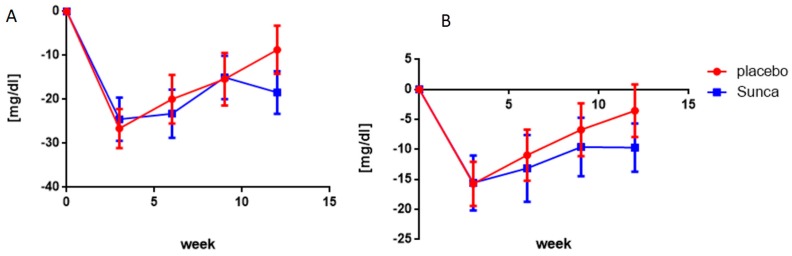
Changes in blood cholesterol (**A**) and LDL-cholesterol (**B**) at various intervals after consumption of either sunflower extract (in blue) or placebo (in red). Each symbol represents the mean ± s.e.m. of at least 17 observations.

**Figure 4 nutrients-11-01080-f004:**
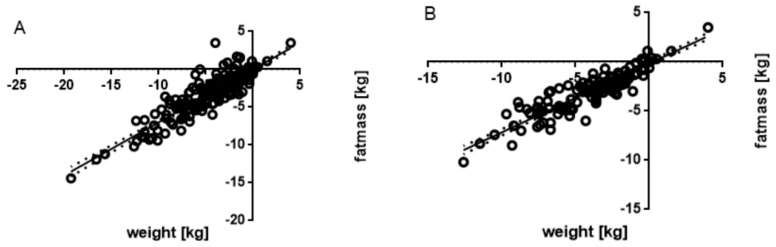
Association between the change in body weight and that in fat mass for all subjects (**A**) and women only (**B**). Each symbol represents one observation. The dotted area indicates the 95% confidence interval of the regression line.

**Table 1 nutrients-11-01080-t001:** Chlorogenic acids content of sunflower extract expressed as 5-CQA equivalent (%).

Compound	Retention Time (min)	%
3-Caffeoyl quinic acid (3-CQA) (**1**)	3.64	5.72
5-Caffeoyl quinic acid (5-CQA) (**2**)	5.33	26.66
4-Caffeoyl quinic acid (4-CQA) (**3**)	5.50	8.51
3,4-Di-caffeoyl quinic acid (3,4-Di CQA) (**4**)	10.76	1.19
3,5-Di-caffeoyl quinic acid (3,5-Di CQA) (**5**)	11.22	1.07
4,5-Di-caffeoyl quinic acid (4,5-Di CQA) (**6**)	12.29	1.47
	Total	44.62

**Table 2 nutrients-11-01080-t002:** Characteristics of the participants at baseline (V0).

Variable (Unit)	Placebo Group (*N* = 20) (5 Men and 15 Women)		Sunflower Extract Group (*N* = 30) (13 Men and 17 Women)		*p* Value (Intergroup)
	Mean ± SD	Range	Mean ± SD	Range	
Age (years)	40.5 ± 9.57	21–51	44.5 ± 9.69	23–64	0.15
Weight (kg)	93.8 ± 11.86	73.8–122.2	101.0 ± 14.46	74.2–129.9	0.07
Height (cm)	165.9 ± 7.58	150–183	168.7 ± 9.91	154–192	0.28
BMI (kg/m^2^)	34.0 ± 2.91	30.4–39.1	35.3 ± 2.93	30.1–39.9	0.12
Waist circumference (cm)	109.1 ± 9.16	95–129	114.0 ± 8.48	95–136	0.06
Lean mass (kg)	55.0 ± 10.09	43.9–84.3	60.9 ± 13.01	44.9–88.5	0.09
Fat mass (kg)	38.8 ± 7.31	24.9–53.6	40.0 ± 7.22	27.5–58.3	0.57
MM/FM (ratio)	1.5 ± 0.47	0.99–2.67	1.6 ± 0.45	0.96–2.61	0.47
Total body water (kg)	40.3 ± 7.39	32.1–61.7	44.1 ± 9.31	32.9–64.8	0.12
Fat (%)	41.4 ± 6.29	27.2–50.0	40.0 ± 6.58	27.7–51.1	0.47
Glucose (mg/dl)	92.9 ± 14.32	73–143	96.5 ± 18.18	76–177	0.37
HbA1c (%)	5.7 ± 0.54	4.7–7.4	5.7 ± 0.55	5.0–7.7	0.83
Cholesterol (mg/dL)	203.7 ± 42.20	131.0–291.0	207.7 ± 41.69	129.0–299.0	0.74
Triglycerides (mg/dL)	116.3 ± 63.53	60.0–331.0	130.1 ± 74.06	33.0–322.0	0.45
LDL (mg/dL)	132.3 ± 33.87	75–216	132.7 ± 35.83	54.0–195.0	0.97
HDL (mg/dL)	48.7 ± 14.31	31.0–96.0	46.7 ± 11.74	26.0–73.0	0.66
LDL/HDL (ratio)	2.7± 0.68	1.8–4.2	2.9 ± 0.97	1.6–5.1	0.46
Free Fatty Acids (nmol/dL)	0.4 ± 0.15	0.24–0.82	0.4 ± 0.15	0.09–0.73	0.90

**Table 3 nutrients-11-01080-t003:** Mean change values ± 1 × standard error of the mean from baseline to final visit (V5–V0) of outcomes parameters.

Variable (Unit)	All Subjects (*N* = 50)			Subjects > 30 Years (*N* = 44)			Women > 30 years (*N* = 30)		
	Placebo (*N* = 18)	Sunflower Extract (*N* = 28)	*p*-Value	Placebo (*N* = 17)	Sunflower Extract (*N* = 26)	*p*-Value	Placebo (*N* = 13)	Sunflower Extract (*N* = 15)	*p*-Value
Anthropometric measurements									
Weight (kg)	−5.53 ± 1.30	−6.90 ± 1.30	0.092	−5.71 ± 1.38	−7.28 ± 1.43	0.043	−4.93 ± 0.68	−6.01 ± 0.94	0.167
BMI (kg/m^2^)	−1.88 ± 0.32	−2.60 ± 0.33	0.011	−1.94 ± 0.33	−2.75 ± 0.33	0.004	−1.81 ± 0.25	−2.31 ± 0.35	0.109
WC (cm)	−4.75 ± 1.76	−8.44 ± 0.81	0.000	−4.91 ± 1.86	−8.82 ± 0.81	0.000	−3.23 ± 2.11	−9.15 ± 0.87	0.000
Bioimpedance analysis									
MM (kg)	−1.54 ± 0.43	−1.88 ± 0.31	0.403	−1.69 ± 0.43	−1.86 ± 0.32	0.265	−1.37 ± 0.42	−1.22 ± 0.37	0.219
FM (kg)	−3.98 ± 0.71	−4.87 ± 0.69	0.095	−4.02 ± 0.75	−5.26 ± 0.68	0.018	−3.56 ± 0.44	−4.72 ± 0.74	0.026
MM/FM (ratio)	0.19 ± 0.06	0.20 ± 0.04	0.543	0.19 ± 0.07	0.22 ± 0.04	0.511	0.09 ± 0.02	0.14 ± 0.03	0.004
TBW (kg)	−1.13 ± 0.32	−0.90 ± 0.52	0.376	−1.24 ± 0.32	−0.85 ± 0.56	0.258	−1.00 ± 0.32	−0.89 ± 0.27	0.157
Fat (%)	−2.03 ± 0.47	−2.44 ± 0.40	0.255	−2.00 ± 0.49	−2.69 ± 0.39	0.047	−1.62 ± 0.31	−2.41 ± 0.44	0.014
Biochemical analysis									
Glucose (mg/dl)	−2.61 ± 2.61	−3.71 ± 1.80	0.132	−2.41 ± 2.76	−4.35 ± 1.88	0.227	−5.00 ± 1.78	−0.53 ± 1.82	0.046
HbA1c (%)	−0.13 ± 0.03	−0.13 ± 0.05	0.664	−0.12 ± 0.03	−0.14 ± 0.06	0.700	−0.11 ± 0.04	−0.07 ± 0.04	0.624
Cholesterol (mg/dL)	−8.72 ± 5.45	−18.43 ± 4.86	0.018	−8.65 ± 5.78	−18.31 ± 5.07	0.028	−4.85 ± 5.97	−15.53 ± 4.76	0.578
Triglycerides (TG) (mg/dL)	−23.94 ± 14.45	−25.46 ± 9.22	0.411	−23.29 ± 15.31	−26.08 ± 9.85	0.342	−6.54 ± 8.53	−25.00 ± 7.33	0.065
LDL (mg/dL)	−3.53 ± 4.37	−9.69 ± 4.02	0.150	−3.69 ± 4.65	−9.71 ± 4.19	0.164	−2.00 ± 4.91	−8.20 ± 4.62	0.350
HDL (mg/dL)	−0.78 ± 2.15	−2.75 ± 0.85	0.327	−0.71 ± 2.28	−2.42 ± 0.88	0.427	−1.69 ± 2.90	−2.27 ± 1.42	0.112
LDL/HDL (ratio)	−0.03 ± 0.12	−0.02 ± 0.10	0.465	−0.04 ± 0.13	− 0.03 ± 0.10	0.433	0.06 ± 0.14	0.00 ± 0.10	0.265
FFA (nmol/dL)	0.14 ± 0.06	0.09 ± 0.05	0.370	0.15 ± 0.06	0.07 ± 0.04	0.310	0.15 ± 0.08	0.04 ± 0.05	0.255

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
