# Peer review of "Helianthus annuus Seed Extract Affects Weight and Body Composition of Healthy Obese Adults during 12 Weeks of Consumption: A Randomized, Double-Blind, Placebo-Controlled Pilot Study"

_nutrients, 2019, doi:10.3390/nu11051080_

Round 1

Reviewer 1 Report

The authors have investigated the anthropometric and anti-lipidemic effects of CGAs from sunflower by Randomized Clinical Trial. They concluded by performing the trial that consumption of 500 mg/day of a 40% CAGs from sunflower for 12 weeks significantly decreased BMIs, body weight and improved fat-related parameters.

This study can not be accepted in its current form. This study needs to explain certain results presented in the manuscript.

1) The figure-3 shows reduction in cholesterol and LDLs. However we see significant decrease in both upon placebo as well. The P value shown here is the comparison between the placebo and the treatment or is it compared to zero? also as you see the effect after day 0, both are sort of parallel going. The authors needs to explain this observation more in detail.

Author Response

We did not modified the introduction considering that it provide sufficient background.

We added details to the methods description.

We clarified the part dealing with the decrease of cholesterol and LDL as required by the reviewer.

With the details added and the clarification concerning the cholesterol and improved explanation about the findings related to the efficacy and gender, the conclusions are better supported by the results.

Reviewer 2 Report

In this pilot study, the authors examined the effects of sunflower extract ingestion for 12 weeks on body weight and composition in obese adults. The results demonstrated that sunflower extract ingestion reduced body weight, which mainly accounted for a decrease in adiposity. This finding is of enough interest in this journal, and the paper was well written; however, there is still room for improvement. Specific comments are listed below.

1) There is no description of clinical trial registration.

2) L 136: The explanation of "hypocaloric diet recommendation" is not enough. The present study did not examine the simple effect of sunflower extract on body fatness; therefore, the possibility that the hypocaloric diet enhanced or suppressed the beneficial role of sunflower extract cannot be denied.

3) Strictly speaking, "muscular mass" used in the manuscript should be replaced with lean mass, which is the sum of muscular mass and bone mass.

4) Table 3 and Figure 3: SEM seems to be better than SD in this case because readers want to focus on the difference between the two groups but not variance in each group.

5) Since the present study was a pilot study and the subanalyses were retrospective, the authors should be more careful not to emphasize the findings. The mechanism why obese female individuals above 30 years old showed higher responsiveness to sunflower extract was not clear.

6) L 110: Please spell out "HPLC."

7) L 298: Please correct "GCAs" to "CGAs."

8) Figure 4: "chweight" and "chfatmass" are not common expressions. 

Author Response

Details dealing with the trial registration have been added.

The explanation dealing with the hypocaloric diet has been added.

Muscular mass has been replaced by lean mas, we agree with the reviewer opinion.

Table 3 and Figure 3 are now presenting the results with SEM as required by the reviewer.

Dealing with the subanalyse we added comments and hypothesis describing why obese female individuals over 30 years old are better responders. As required by the reviewer.

HPLC has been speel out.

GCAs has been corrected in CGAs.

Figure 4 has been corrected as indicated by the reviewer.

We added details to better description of the methods.

We improved the results presentation concerening the cholesterol and LDL decrease.